# Identification of New Genetic Determinants in Pediatric Patients with Familial Hypercholesterolemia Using a Custom NGS Panel

**DOI:** 10.3390/genes13060999

**Published:** 2022-06-01

**Authors:** Lena Rutkowska, Kinga Sałacińska, Dominik Salachna, Paweł Matusik, Iwona Pinkier, Łukasz Kępczyński, Małgorzata Piotrowicz, Ewa Starostecka, Andrzej Lewiński, Agnieszka Gach

**Affiliations:** 1Department of Genetics, Polish Mother’s Memorial Hospital—Research Institute, 93-338 Lodz, Poland; kinga.salacinska.lodz@gmail.com (K.S.); dominik.salachna@iczmp.edu.pl (D.S.); iwona.pin@interia.pl (I.P.); lukasz.kepczynski@iczmp.edu.pl (Ł.K.); mpmag@wp.pl (M.P.); 2Department of Pediatrics, Pediatric Obesity and Metabolic Bone Diseases, Faculty of Medical Sciences in Katowice, Medical University of Silesia, 40-055 Katowice, Poland; endocrin@wp.pl; 3Department of Endocrinology and Metabolic Disease, Polish Mother’s Memorial Hospital—Research Institute, 93-338 Lodz, Poland; ewastarostecka@wp.pl; 4Department of Endocrinology and Metabolic Diseases, Medical University of Lodz, 90-419 Lodz, Poland

**Keywords:** familial hypercholesterolemia, genetics, next generation sequencing, *LDLR* gene, dyslipidemia, novel variant, pediatric patients

## Abstract

The most common form of inherited lipid disorders is familial hypercholesterolemia (FH). It is characterized primarily by high concentrations of the clinical triad of low-density lipoprotein cholesterol, tendon xanthomas and premature CVD. The well-known genetic background are mutations in *LDLR*, *APOB* and *PCSK9* gene. Causative mutations can be found in 60–80% of definite FH patients and 20–30% of those with possible FH. Their occurrence could be attributed to the activity of minor candidate genes, whose causal mechanism has not been fully discovered. The aim of the conducted study was to identify disease-causing mutations in FH-related and candidate genes in pediatric patients from Poland using next generation sequencing (NGS). An NGS custom panel was designed to cover 21 causative and candidate genes linked to primary dyslipidemia. Recruitment was performed using Simon Broome diagnostic criteria. Targeted next generation sequencing was performed on a MiniSeq sequencer (Illumina, San Diego, CA, USA) using a 2 × 150 bp paired-end read module. Sequencing data analysis revealed pathogenic and possibly pathogenic variants in 33 out of 57 studied children. The affected genes were *LDLR*, *APOB*, *ABCG5* and *LPL*. A novel pathogenic 7bp frameshift deletion c.373_379delCAGTTCG in the exon 4 of the *LDLR* gene was found. Our findings are the first to identify the c.373_379delCAGTTCG mutation in the *LDLR* gene. Furthermore, the double heterozygous carrier of frameshift insertion c.2416dupG in the *LDLR* gene and missense variant c.10708C>T in the *APOB* gene was identified. The c.2416dupG variant was defined as pathogenic, as confirmed by its cosegregation with hypercholesterolemia in the proband’s family. Although the *APOB* c.10708C>T variant was previously detected in hypercholesterolemic patients, our data seem to demonstrate no clinical impact. Two missense variants in the *LPL* gene associated with elevated triglyceride plasma level (c.106G>A and c.953A>G) were also identified. The custom NGS panel proved to be an effective research tool for identifying new causative aberrations in a genetically heterogeneous disease as familial hypercholesterolemia (FH). Our findings expand the spectrum of variants associated with the FH loci and will be of value in genetic counseling among patients with the disease.

## 1. Introduction

Cardiovascular diseases (CVDs) remain the leading cause of death worldwide, with an estimated 17.9 million deaths attributed to cardiovascular incidents each year [1]. The underlying cause of CVD is the continual deposition of lipoproteins and calcium in the arterial intima, resulting in the development of inflammation and subsequent fibrosis, called arteriosclerosis. Although the clinical complications of atherosclerosis usually present in middle age, it is well known that atherosclerotic plaque may start to build up at a young age; indeed, reports exist of fatty streaks in the aortas and coronary arteries in patients aged only 10 years [2]. Efficient diagnosis of dyslipidemia and the implementation of early treatment in children may reduce their risk of accelerated atherosclerosis and premature CVD. A key role in the development of atherosclerosis in children is played by inherited factors known to cause high blood lipid levels; it is believed that such familial hypercholesterolemia (FH) is present in 1 in 31 individuals with atherosclerotic cardiovascular disease (ASCVD), especially those with premature ASCVD (around 1 in 15 individuals). No other single disorder is responsible for so many ASCVD patients [3].

Familial hypercholesterolemia (FH) constitutes the most common form of inherited dyslipidemia. It is mainly characterized by high concentrations of the clinical triad of low-density lipoprotein (LDL) cholesterol, tendon xanthomas and premature CVD. Atherogenic cholesterol-rich LDLs are usually removed from circulation in the liver; however, any hepatic dysfunction results in their accumulation in the artery wall [4]. Their deposition in the artery stimulates an inflammatory response, which in turn results in damage to the wall and the formation of atherosclerotic plaques: coronary calcification is present in about 25% of 11- to 23-year-old children and young adults with heterozygous FH [4].

The prevalence of the heterozygous form (HeFH) is estimated to be 1:500, but recent studies suggest that it could be as high as 1:250 to 1:380 [5]. The most severe form of FH is homozygous familial hypercholesterolemia (HoFH). Similarly to HeFH, the incidence of HoFH is greater than previously thought, being currently estimated at 1:160,000–300,000 [6], this greater prevalence could be due to a combination of increased awareness of the disease and the rapid development of genetic testing. The genetic background of FH includes mutations in *LDLR* (low-density lipoprotein receptor gene), *APOB* (apolipoprotein B gene) and *PCSK9* (proprotein convertase subtilisin/kexin type 9 gene).

However, in most cases, hypercholesterolemia is caused by a loss-of-function mutation in *LDLR*, and over 2000 functional mutations have been documented so far in this gene [5]. The low-density lipoprotein receptor (LDLR) gene family consists of cell surface proteins involved in the receptor-mediated endocytosis of specific ligands. LDLR plays a critical role in the homeostatic control of blood cholesterol by mediating its removal from circulation. Lipoproteins are bound to the LDL receptor and taken into the lysosomes to be degraded, following which the unoccupied receptor cycles back to the cell surface. Plasma LDL is then eliminated primarily via hepatic LDLR [7]. The various stages involved in the posttranslational processing, binding, uptake and subsequent dissociation of the LDL particle-receptor complex are subject to disruption by mutations in the *LDLR* gene, resulting in the development of lipid disorders [8].

The LDL receptor also captures and binds the apolipoprotein B and apolipoprotein E; lipoproteins are responsible for controlling the cellular uptake of lipoproteins by carrying lipids, including cholesterol. The presence of *APOB* or *APOE* mutations results in structural rearrangements within apolipoprotein domains, resulting in improper lipid binding and, thus, high blood lipid levels. The protein that regulates the exposure of the LDL receptor on the hepatocyte surface and thus regulates LDL particle uptake is PCSK9. Circulating PCSK9 binds to LDLR and targets it for lysosomal degradation. Both *PCSK9* and *LDLR* gene expression is coregulated at the transcriptional level by *SREBP-2* (sterol regulatory element-binding protein-2) [9]. Gain-of-function mutations in *PCSK9* gene lead to autosomal dominant hypercholesterolemia, while loss-of-function mutations are associated with reduced LDL-C plasma levels [7]. In the last decade, PCSK9 inhibitors have become a major drug target in cardiovascular medicine. In addition, a broad range of minor potential FH genes are continually being identified, such as *STAP1* (signal-transducing adaptor family member 1), which has been proposed as a fourth causative gene for familial hypercholesterolemia, even though the molecular mechanisms by which the adapter protein could affect cholesterol metabolism remain unclear [10,11].

Most FH patients are known to carry autosomal dominant mutations in the *LDLR*, *APOB* or *PCSK9* gene. Despite this, some patients also present a rare autosomal recessive form of the disease caused by a homozygous or compound heterozygous mutation in the *LDLRAP1* gene. The LDLRAP1 protein binds to LDLR and allows the internalization of the LDL-LDLR complex on the hepatocyte surface. The resulting molecular defects in LDLRAP1 may impair the LDL-cholesterol metabolism by causing a severe reduction of LDL uptake [12].

The next core component that plays a key role in the lipoprotein metabolism is lipoprotein lipase (LPL), encoded by the *LPL* gene. This enzyme hydrolyses the triglycerides carried in chylomicrons and VLDL to fatty acids, which can be taken up by cells. Alterations in the *LPL* gene contribute to severe hypertriglyceridemia. The activity of the LPL enzyme is regulated by four apolipoproteins: ApoC1, ApoC2, ApoC3 and ApoA5. Apolipoprotein C2 serves as a cofactor, while ApoA5 probably works as an LPL activator. Alterations in the *APOC2* or *APOA5* genes can potentially affect enzyme activity. In addition, proper LPL functioning requires the activity of GPIHBP1, expressed on capillary endothelial cells. Its primary function is to bind LPL in the subendothelial spaces and transport it to the capillary lumen. Various *GPIHBP1* mutations have been found to prevent GPIHBP1 from binding LPL, resulting in invalid processing of TG and severe hypertriglyceridemia [13].

It is estimated that FH-causing mutations can be found in 60–80% of definite FH patients and 20–30% of those with possible FH [14]. However, in other pediatric cases, the diagnosis can be complicated by the presence of a polygenic background or mutations in one of many candidate genes related to primary dyslipidemia. Next-generation sequencing (NGS) has presented an opportunity to identify additional ultra-rare causes of FH, including atypical clinical manifestations resulting from rare mutations in the *APOA5*, *STAP1* or *LIPA* genes. Targeted NGS allows massive parallel sequencing to be performed on regions of interest, providing a wide range of obtained data to better understand the heterogeneity of familial hypercholesterolemia.

Several different types of clinical diagnostic criteria have been established for FH globally, such as Dutch Lipid Clinical Network (DLCN), Make Early Diagnosis to Prevent Early Deaths (MedPed) diagnostic criteria or Simon Broome diagnostic criteria [3]. The Simon Broome diagnostic criteria and DLCN are generally based on similar assumptions, such as elevated cholesterol level, clinical characteristics, family history of CVD and molecular diagnosis. The MedPed classification is based on plasma cholesterol level and is dependent on patient age and family history [15]. As characteristic phenotype indicators are not revealed until a certain age in most pediatric patients, the clinical diagnosis is often based on lipid profile and a family history of CVD. Furthermore, as LDL cholesterol concentrations fluctuate greatly during adolescence, a mild course may go unnoticed. Childhood is the optimal period for FH diagnosis, due to the minimal dietary or hormonal impact, and all suspected individuals should be promptly referred to genetic tasting. An accurate confirmatory diagnosis can greatly increase the compliance of long-term preventive therapy, especially in pediatric patients.

The aim of the study was to identify known and new genetic variants in pediatric patients with familial hypercholesterolemia using a custom NGS panel. Our hypothesis assumes that the genetic background of familial hypercholesterolemia in children is highly heterogeneous and NGS is thus a suitable tool for diagnosing it.

## 2. Materials and Methods

Genomic DNA was isolated from peripheral blood samples using a MagCore automatic nucleic acid extractor (RBC Bioscience, New Taipei City, Taiwan). The quantitative and qualitative assessment of extracted DNA was performed on a NanoDrop 2000 spectrophotometer (Thermo Fisher Scientific, Waltham, MA, USA). In accordance with the Illumina protocol, the samples were measured using a Quantus Fluorometer (Promega, Madison, WI, USA) and diluted to the required concentration (10 ng/µL). The library preparation procedure was conducted in accordance with Illumina TruSeq Custom Amplicon Low Input Library Prep Reference Guide. Hybridization, ligation and PCR protocol were carried out with a T100 Thermal Cycler (Bio-Rad, Hercules, CA, USA), as recommended. The indexed samples were pooled, diluted and combined with 1% PhiX control. Targeted next generation sequencing (NGS) was performed on a MiniSeq sequencer (Illumina, San Diego, CA, USA) using a 2 × 150 bp paired-end read module. The minimum reading depth was 100-fold. A custom NGS panel containing 21 causative and candidate genes linked to familial hypercholesterolemia and other primary dyslipidemias was designed using Illumina DesignStudio software. The following genes included: *ABCA1*, *ABCG5*, *ABCG8*, *APOA5*, *APOB*, *APOC2*, *APOE*, *CYP7A1*, *GPIHBP1*, *LCAT*, *LDLR*, *LDLRAP1*, *LIPA*, *LMF1*, *LMNA*, *LPL*, *PCSK9*, *PPARG*, *SCAP*, *SREBF2*, *STAP1*.

The obtained NGS data were processed and analyzed by VariantStudio Software, compatible with the Illumina platform. The pathogenicity of the variants was determined in silico using web-based software, such as PolyPhen2, SIFT and Mutation Taster. Searches for phenotype–genotype correlations were evaluated using PubMed, LOVD or VARSOME databases. Variants were classified according to current American College of Medical Genetics and Genomics (ACMG) guidelines [16].

The presence of selected variants was confirmed by bidirectional Sanger sequencing on a 3500 Series Genetic Analyzer (Applied Biosystems, Waltham, MA, USA). DNA Variant Analysis was performed using Mutation Surveyor V5.1.0 software (SoftGenetics, State College, PA, USA).

The inclusion criteria for the study were the Simon Broome diagnostic criteria. The most commonly used are the Dutch Lipid Clinical Network Criteria, but those cannot be applied to pediatric patients. According to the Simon Broome diagnostic criteria, definite familial hypercholesterolemia occurs when a child below 16 years of age has a total cholesterol level greater than 260 mg/dL or an LDL-cholesterol above 155 mg/dL. Furthermore, it is necessary to confirm the presence of tendon xanthomas in the proband child or their first/second degree relative or DNA-based evidence of an *LDLR*, *APOB* or *PCSK9* mutation. In a child under 16 years, familial hypercholesterolemia can be suspected in cases characterized by equivalent LDL-C and total cholesterol levels, without tendon xanthomas or a DNA-based test but with a positive family history, i.e., presence of myocardial infarction in relatives or total cholesterol level greater than 290 mg/dL in adult relatives or 260 mg/dL in relatives under 16 years. For children and adults over 16 years old, the lipid cut-offs are TC greater than 290 mg/dL or LDL-cholesterol above 190 mg/dL.

The study was conducted in accordance with the Declaration of Helsinki and approved by the Ethics Committee of the Polish Mother’s Memorial Hospital—Research Institute (opinion number 15/2016, date of approval 12 January 2016). Informed consent was obtained from all subjects involved in the study. The entire study was performed in the Department of Genetics, Polish Mother’s Memorial Hospital—Research Institute, Lodz, Poland.

## 3. Results

A group of 57 children with a clinical suspicion of familial hypercholesterolemia (FH) was recruited for the present study. The children ranged in age from 2 to 17 years (mean age: 10). In the study group, 48 patients fulfilled the Simon Broome criteria for lipids, of which 33 presented pathogenic aberrations. Among the eight participants not meeting the criteria, only one obtained a positive genetic result. Patient age, lipid profile and obtained diagnostic results are shown in Table 1. Sibling cases are labelled with the same number with an ‘A’ and ‘B’ designation.

Table 2 compares mean total cholesterol (TC), LDL-cholesterol, HDL-cholesterol and triglyceride (TG) content, as well as the standard deviation, for whole study group with the values for two separate subgroups of children, with or without genetic findings.

Sequencing data analysis revealed possibly pathogenic variants in 33 out of 57 pediatric cases. Among the seventeen presumed pathogenic variants, listed in Table 3, twelve were missense, three frameshift and two splice site mutations. In addition, three variants of uncertain significance and one risk factor were also detected. The affected genes were *LDLR*, *APOB*, *ABCG5* and *LPL*. No potential pathogenic variants were found within the *ABCA1*, *ABCG8*, *APOA5*, *APOC2*, *APOE*, *CYP7A1*, *GPIHBP1*, *LCAT*, *LDLRAP1*, *LIPA*, *LMF1*, *LMNA*, *PCSK9*, *PPARG*, *SCAP*, *SREBF2* of the *STAP1* genes in the pediatric group.

### 3.1. LDLR Gene

Fifteen of the detected pathogenic mutations were located within the *LDLR* gene (c.190+1G>A, c.284G>T, c.373_379delCAGTTCG, c.324_325delGTinsTC, c.530C>T, c.798T>A, c.907C>T, c.1061A>G, c.1433G>A, c.1775G>A, c.1747C>T, c.1846-2A>C, c.1916T>A, c.2063delA, c.2416dupG). The affected exons of *LDLR* were 3–6, 8, 10, 12–14, 17. The gene structure with the positions of the variants are presented in Figure 1.

The most common variant in the *LDLR* gene was the heterozygous c.1775G>A p.(Gly592Glu), detected in nine children. In addition, two splice site variants c.190+1G>A and c.1846-2A>C were detected: the former in two patients and the latter in one. The first case was a 12-year-old boy (patient 8) with a total cholesterol level of 318 mg/dL and LDL cholesterol level of 237 mg/dL. The boy was found to harbor the c.190+1G>A mutation. Sanger sequencing revealed the occurrence of the same mutation in the mother, who was symptomatic, but not in the father, who was normolipemic.

The second case was a five-year-old girl (patient 31) with severe hypercholesterolemia (TC 345 mg/dL and LDL-c 273 mg/dL) at the time of referral to genetic counselling. Her cholesterol level after the first year of life was approximately 500 mg/dL. The girl is being treated with monacolin K. Sanger sequencing confirmed the occurrence of the same mutation in the mother, who also suffered from hypercholesterolemia (TC 320 mg/dL and LDL-c 233 mg/dL). During Ezetimibum treatment, the TC level fell to 251 mg/dL and LDL-c to 167 mg/dL.

In a third case, the splice site variant c.1846-2A>C was identified in a nine-year-old girl (patient 5) with a TC of 312 mg/dL and an LDL-c of 248 mg/dL. Both mutations had previously been reported as disease-causing in a Polish study of familial hypercholesterolemia [17].

NGS analysis revealed the presence of a novel 7bp frameshift deletion c.373_379delCAGTTCG in the exon 4 of the *LDLR* gene in two siblings. The lipid parameters of the elder, 10-year-old, brother (patient 7b) showed severe hypercholesterolemia with significantly elevated total cholesterol and LDL cholesterol levels, these being 426 mg/dL and 348 mg/dL, respectively. The younger, five-year-old, sister (patient 7a) also demonstrated elevated TC (278 mg/dL) and LDL cholesterol levels (170 mg/dL). The performed in silico analysis classified the c.373_379delCAGTTCG variant as possibly pathogenic. The predicted consequences of 7bp deletion, with Gln125Ser at the first amino acid change, is the formation of a defective transcript and induction of a nonsense-mediated mRNA decay mechanism (NMD). Looking more closely at the deleted nucleotides and their flanking sequences, 3 bp directed repeats can be observed (Figure 2). The cause of the aberration may be the presence of specific deletion and repeat localizations, as described in the slipped mispairing hypothesis [18].

In the siblings, Sanger sequencing confirmed the presence of the c.373_379delCAGTTCG variant (Figure 3), which was found to be of paternal origin. The variant cosegregates with elevated TC and LDL in the proband’s family (Table 4), which supports the evidence of variant pathogenicity. This is the first report of a c.373_379delCAGTTCG mutation in the *LDLR* gene. The variant was submitted to ClinVar and was assigned with the accession number VCV001300030.1.

The next interesting case was that of a 16-year-old girl (patient 6) who was found to be a carrier of a frameshift insertion c.2416dupG in the exon 17 of the *LDLR* gene and missense variant c.10708C>T in the exon 26 of the *APOB* gene. Firstly, the c.2416dupG mutation causes a shift in the reading frame at codon 806, changing valine into glycine, thus creating a premature stop codon at a new 11 position. This mutation has previously been described as likely pathogenic. The examination of the proband’s family confirmed the cosegregation of c.2416dupG with hypercholesterolemia, which may confirm variant pathogenicity (Figure 4).

The second detected SNV c.10708C>T in the *APOB* gene results in histidine to tyrosine amino acid substitution, as previously detected in hypercholesterolemic patients [19]. Prediction tools classified the detected aberration as likely pathogenic. Three members of the described family are c.10708C>T heterozygous carriers, yet their lipid levels are within normal ranges or only slightly elevated. The presence of both DNA variants was confirmed by direct Sanger sequencing (Figure 5).

### 3.2. Other Affected Genes

Searching for possibly pathogenic variants within *APOB* revealed the presence of a c.10580G>A p.(Arg3527Gln) mutation occurring in three pediatric patients. Arg3527Gln is the most frequent alteration within *APOB* in the Caucasian population and is known to have a negative impact on LDL-cholesterol metabolism [20]. The highly conserved receptor binding site is known to be stabilized by the interaction of Arg3527 with Trp4396; as such, the replacement of arginine by a glutamine at position 3527 impairs receptor recognition. The belt conformation of ApoB100 that surrounds the LDL particle is maintained by the interaction of Arg3527 with Trp439615, which stabilizes two clusters of basic amino acids, ensuring the binding of ApoB100 to LDLR [21].

Within the *LPL* gene, two variants were identified: c.106G>A p.(Asp36Asn) and c.953A>G p.(Asn318Ser) (Figure 6). The gene structure with the variant positions are presented in Figure 6.

Despite the discrepancies in the results of predictive programs, it has been proven that the replacement of Asp36Asn results in decreased lipoprotein lipase activity in about 20% of cases [22]. Asp36Asn carriers typically demonstrate hypertriglyceridemia, low HDL level and increased risk of cardiovascular disease [23]. However, contrary to predictions, one 10-year-old boy in the study (patient 14) carrying the c.106G>A mutation, did not present a high triglyceride level (138 mg/dL) but did demonstrate hypercholesterolemia with a TC concentration of 279 mg/dL and an LDL-cholesterol of 158 mg/dL. Sanger sequencing of the mutated sequence in the proband’s parents revealed the presence of c.106G>A in the father (Figure 7). Testing the lipid profile in the father revealed high triglyceride level (314 mg/dL) and total cholesterol level (251 mg/dL), which seems to confirm the expected deleterious effect of Asp36Asn.

The second common variant in the *LPL* gene, c.953A>G p.(Asn318Ser), was detected in a double heterozygous form, together with the missense c.1775G>A p.(Gly592Glu) variant in the *LDLR* gene. This was observed in a 14-year-old boy in the study (patient 32), whose lipid parameters were 320 mg/dL TC and 263 mg/dL LDL-C. The patient received treatment with 5 mg Rosuvastatin, which was found to reduce total cholesterol level to 195 mg/dL and LDL-c level to 135 mg/dL. Variant c.953A>G in the *LPL* gene is frequently reported in patients with familial combined hyperlipidemia and is therefore identified as pathogenic [24].

The data analysis also revealed two possibly pathogenic mutations, c.593G>A and c.1285G>A, within the *ABCG5* gene in two unrelated individuals (patients 3 and 10). However, as no other *ABCG5* or *ABCG8* mutations were identified, the presence of sitosterolemia (autosomal recessive inheritance) was rejected.

## 4. Discussion

The aim of the study was to identify potentially pathogenic FH-related variants in known and candidate genes in pediatric patients, using next generation sequencing (NGS). A group of 57 patients fulfilling the research inclusion criteria was recruited, with a mean age of 10 years. The National Lipid Association Expert Panel on Familial Hypercholesterolemia recommends that the lipid profile should be tested in all children from 9 to 11 years old [25]. However, the only country to fully implement universal screening for early hypercholesterolemia is Slovenia, where testing is carried out in children at the age of five to six years [4]. Their screening algorithm requires direct referral to genetic testing when total cholesterol level is 5–6 mmol/L (193–232 mg/dL) with positive family history, or TC more than 6 mmol/L (232 mg/dL). The Slovenian screening strategy results disease-causing mutations being identified in approximately 45% of tested children [26].

Similar findings were obtained in the present study, with more than half of the examined patients obtaining a positive molecular result. In accordance with Simon Broome threshold lipid values (TC > 260 mg/dL or LDL-cholesterol > 155 mg/dL), 48 recruited patients fulfilled lipid inclusion criteria; of these, 33 obtained a positive genetic result. In the other cases, hypercholesterolemia could be due to the presence of a deep intronic mutation, copy number variation, an aberration in other genes not included in the panel or perhaps the cause may be polygenic. Nevertheless, it seems that the presence of a specific FH phenotype is a more discriminating factor in pediatric patients than in adult patients; another study conducted on a group of adult patients with suspected FH failed to yield such positive genetic findings. In children, the improved recognition of FH could be associated with the lower impact of environmental factors, diet, hormones or comorbidities.

Our approach was based on next generation sequencing (NGS) with the use of a custom designed, 21-gene panel containing well-known FH-related genes (*LDLR*, *APOB* and *PCSK9*) and candidates correlated with lipid disorders, such as *STAP1*, *SCAP*, *LIPA* or *APOC2*. The NGS approach enabled the wide-range analysis of all exons and exon–intron boundaries of the selected genes. The bioinformatic estimation of the obtained results revealed fifteen presumed pathogenic variants in the *LDLR* gene, one in the *APOB* gene and one in the *LPL* gene. The low number of affected genes was probably a consequence of the small study group, particularly since FH is known to demonstrate significant genetic heterogeneity. The predominance of changes within the *LDLR* gene is in line with other FH studies in the Polish population, where pathogenic *LDLR* mutations were reported in 81.4% or 87.1% of all found mutations [14,17].

The most frequent aberration was the missense c.1775G>A (p.Gly592Glu) variant located in the exon 12 of the *LDLR* gene, which was detected in nine pediatric patients. Chmara et al. report that 15% of diagnosed FH cases had the c.1775G>A mutation [20]. It is also one of the most common variants identified in Slovakia and the Czech Republic, with the respective prevalence values of 10.5% and 19.3% [27].

The described aberration is located within a 400-amino acid sequence, formed from three cysteine-rich repeats, in the EGF precursor homology domain [28]. The localization of a highly-conserved glycine residue, within the fifth repeat of “YWTD”, determines its pathogenicity [29]. A Polish FH study found that c.1775G>A, as a LOF mutation, reduces LDL receptor activity to 55–15% of wild type values [20]. A corresponding study from southern Italy identified 89% of LDLR activity in an Italian patient carrying a heterozygous mutation [30]. Other data indicates that the presence of the c.1775G>A mutation results in only 5–15% of LDLR activity [31].

Some discrepancies exist in variant classification. Each *LDLR* mutation can be assigned to one of five functional groups based on the characteristics of the mutant protein [18]. Most studies classified c.1775G>A as Group 5, i.e., a defective recycling of the LDLR protein [29,32,33]. On the other hand, some authors propose Group 2b, i.e., partially disturbed LDLR protein transport from endoplasmic reticulum to the Golgi apparatus or the plasma membrane [20,34,35]. Aside from differences in determining residual LDLR activity level or the dysregulation mechanism of the LDLR pathway, c.1775G>A p.(Gly592Glu) was declared to be pathogenic.

Our findings also include the novel identification of a c.373_379delCAGTTCG (p.Gln125SerfsTer79) frameshift deletion in the exon 4 of the *LDLR* gene. In silico analysis performed by prediction bioinformatics tools indicated the c.373_379delCAGTTCG variant to be pathogenic, potentially shifting the reading frame and therefore generating a premature termination codon at a new 79 position. However, this aberration is not included in any human genome variant databases (HGMD, ClinVar or LOVD) nor any population databases (ExAC, gnomAD or 1000 Genomes Project). Family data appear to confirm variant cosegregation, with significantly elevated LDL cholesterol levels.

There are many hypothesis addressing the mutagenesis of mechanisms leading to microdeletions (<20 bp), one of them being the model of slipped mispairing during the DNA replication process [36]. This mechanism is based on the assumption that 2–8 bp repeats are found in close proximity on complementary DNA strands [37]. Illegitimate pairing between different located repeats during replication leads to single-strand loop formation, which are then removed by DNA repair enzymes. Consequently, the final replication products comprise one deleted and one wild-type DNA duplex [36]. A closer look at the detected c.373_379delCAGTTCG and its contiguous sequence confirmed the presence of those 3 bp repeats. Furthermore, one of the repeated sequence has been removed, suggesting that the slipped mispairing hypothesis is more likely than homologous recombination between palindromic or symmetric repeated sequences [18].

Krawczak et al. report that in 93% of 60 examples of causal human gene deletion of 20 bp or less, the deleted sequence includes or overlaps a direct repeat and that 3 bp repeats were the most frequent [36]. Other studies suggest that 48% of deletional events occur as a consequence of the presence of a repeated flanking motif, caused by a slipped mispairing mechanism. The same authors note that mean LDL receptor activity is reduced to 19.8% among heterozygous carriers of a protein-changing mutation (nonsense, frameshift and splice site mutation) [18].

Similar intragenic *LDLR* deletion has been identified in a Japanese investigation. The study reports that one 7 bp deletion of nucleotides 578–584 in exon 4 was detected among 13 families demonstrating FH. In this case, a 63-year-old patient, the carrier of the aberration, presented corneal arcus, tuberous and Achilles tendon xanthomas on both legs and ischemic heart disease. His total cholesterol concentration was 336 mg/dL, with an LDL cholesterol level of 285 mg/dL. HDL cholesterol and triglyceride levels were within normal ranges. Even though the functional array analysis did not confirm reduced LDL receptor binding activity, the authors propose the existence of other dysfunctional mechanisms, and indicate the aberration as disease-causing. They also assigned the variant to Group 1 mutation (null alleles) [38]. As Group 1 mutations disrupt LDLR synthesis, they generate the most severe phenotypes of the five classes. This group includes mainly nonsense (66.3%) and frameshift mutations (30.4%) [18].

To summarize, the exact pathogenicity mechanisms responsible for the deletion of few bp within the coding region of *LDLR* gene remain generally unclear in most cases. Nevertheless, all authors agree that this type of aberration results in serious disturbances in cholesterol metabolism. Based on family examination, the available literature data and estimation by predictive programs, the detected novel c.373_379delCAGTTCG variant was considered to be pathogenic.

Our findings also reveal the presence of a double-heterozygous carrier, with the frameshift mutation c.2416dupG in the exon 17 of the *LDLR* gene and the missense variant c.10708C>T in the exon 26 of the *APOB* gene. Exon 17 of the *LDLR* gene encodes membrane-anchoring and cytoplasmic tail domains, which are essential for ensuring the cell membrane attachment of the receptor and correct protein endocytosis [18,39]. The c.2416dupG variant may result in nonsense-mediated mRNA decay (NMD), which is a translation-coupled mechanism that eliminates mRNAs containing premature translation-termination codons or protein truncation. In each case, the function is severely disturbed. Previous studies have reported this aberration in FH patients in the US, southern Sweden, the Czech Republic, the Netherlands and Japan [39]. Our identified family histories (Figure 4) indicate that the c.2416dupG variant cosegregates with a prominent FH-phenotype, which supports its pathogenicity. The second c.10708C>T, p.(His3570Tyr) variant in the *APOB* gene has previously been described as a potentially FH-related mutation, even though its pathogenicity is still unproven [19,40,41]. Recent studies suggest that the p.His3570Tyr variant has a minor effect on LDL-C levels and would be better classified as a polymorphism in the *APOB* gene. Other authors propose that the rare c.10708C>T variant alone is not pathogenic but that its co-occurrence with *LDLR* mutation can lead to a more severe phenotype in terms of atherosclerotic vascular disease [42]. Our data demonstrate that there is no evidence that *APOB* c.10708C>T has a causative effect; the variant also occurs in unaffected adult family members.

The second double-heterozygous patient was a 14-year-old boy carrying the common c.953A>G variant in *LPL* gene and the c.1775G>A variant in the *LDLR* gene. His lipid profile showed isolated hypercholesterolemia, which clearly demonstrates the pathogenic effect of the *LDLR* variant. However, the negative influence of the second variant within the *LPL* gene is not so apparent. Population-based studies indicate that the frequency of the heterozygous c.953A>G (p.Asn318Ser) variant ranges from 1% to 7% among control individuals, and that heterozygous carriers demonstrated a 31% mean increase in plasma triglyceride level [43]. The impact of c.953A>G in our case is questionable due to a lack of any visible effect in the patient lipidogram. Perhaps functional testing would be conclusive in this case.

The next-generation sequencing proved to be an appropriate tool for identifying changes in FH pediatric patients. Of course, while the choice of NGS over WES may be considered as a main study limitation, the substantially lower cost of NGS, with WES being three times as expensive, makes the NGS technology more achievable for Polish research institutions. Reducing the scope of sequencing to exons of selected genes narrows the diagnostic possibilities, but the resulting reduction in costs allows more patients to be tested. Moreover, panel sequencing platforms are a more preferred choice in Poland, due to reduced equipment and service costs and the greater availability of service facilities and specialists.

## 5. Conclusions

Our findings provide a greater insight into the genetic complexity of primary dyslipidemias. Despite the relatively limited size of the study group, a wide variety of changes were noted within the *LDLR* gene. This best illustrates the pressing need for the development of new diagnostic approaches in the field of genetic testing. More powerful tools, such as next generation sequencing, increase the chance of accurate diagnosis and thus the implementation of early preventive treatment.

## Figures and Tables

**Figure 1 genes-13-00999-f001:**
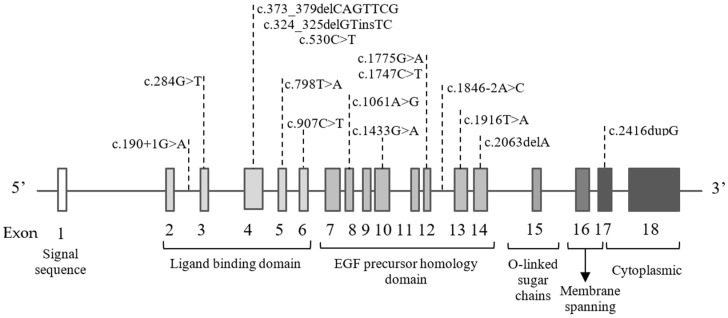
Schematic presentation of eighteen exons and six corresponding functional domains in the *LDLR* gene with the positions of the variants indicated.

**Figure 2 genes-13-00999-f002:**
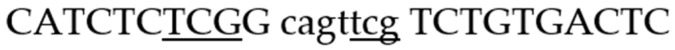
10 bp sequences bordering the deleted nucleotides (small letters). Underlining indicates the direct repeats.

**Figure 3 genes-13-00999-f003:**
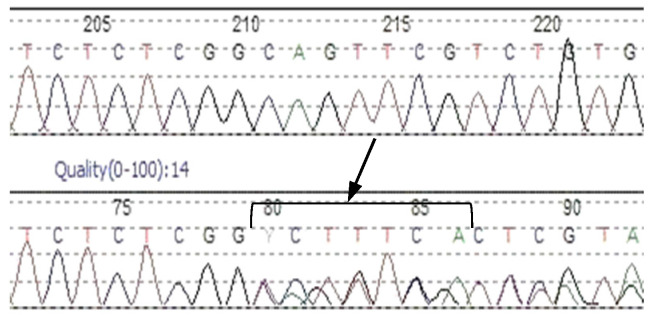
Sanger sequencing chromatogram confirms a heterozygous c.373_379delCAGTTCG variant in the exon 4 of the *LDLR* gene.

**Figure 4 genes-13-00999-f004:**
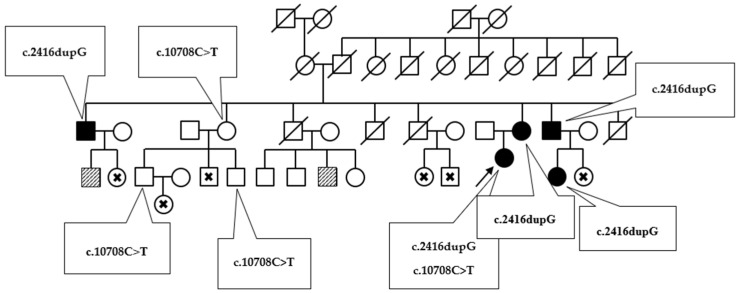
The diagram presents the inheritance of variants c.2416dupG and c.10708C>T in the proband’s family. Phenotypically affected members are marked by black. Patients with slightly elevated total cholesterol level are flagged by a diagonal slash. Examined patients with no clinical symptoms are marked with a cross.

**Figure 5 genes-13-00999-f005:**
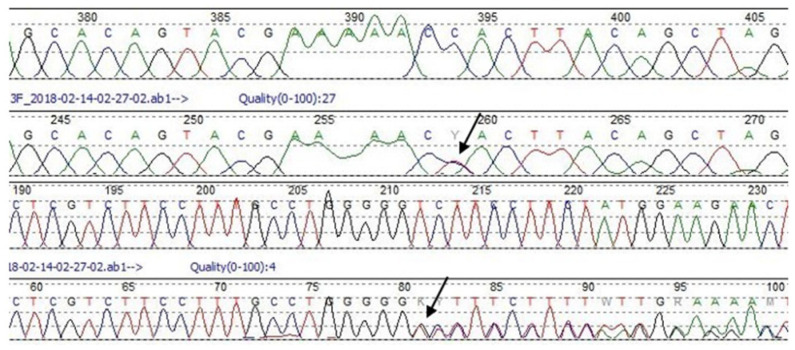
Sanger sequencing chromatograms showing heterozygous c.2416dupG variant in the exon 17 of the *LDLR* gene (upper chromatogram) and heterozygous c.10708C>T variant in the exon 26 of the *APOB* gene (lower chromatogram). The top lane picks refer to wild type sequence. The middle lane picks present described aberrations.

**Figure 6 genes-13-00999-f006:**
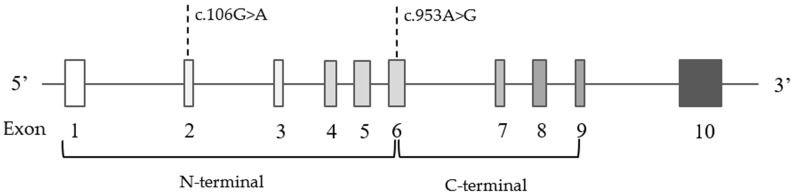
Schematic presentation of ten exons and corresponding two functional domains in the *LPL* gene with indicated variant positions.

**Figure 7 genes-13-00999-f007:**
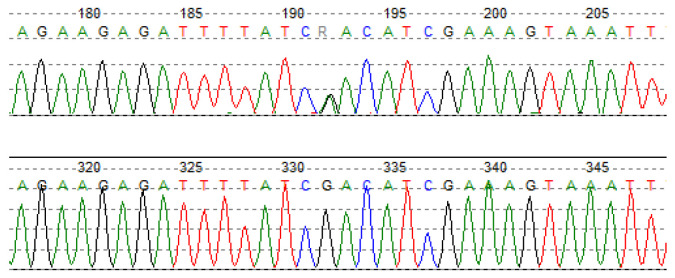
Sanger sequencing chromatogram shows heterozygous c.106G>A variant in the exon 2 of the *LPL* gene.

**Table 1 genes-13-00999-t001:** Patient data and obtained genetic results.

Patient	Age	Sex	TC [mg/dL]	LDL-C [mg/dL]	HDL-C [mg/dL]	TG [mg/dL]	NGS Result	
							HGVSc	Gene
1.	15	M	293	253	42	45	-	
2A.	6	F	228	152	NA	47	-	
2B.	10	F	226	155	51	98	-	
3.	11	M	228	134	55	190	NM_022436.2:c.593G>A	*ABCG5*
4.	3	F	277	222	47	43	-	
5.	9	F	312	248	37	137	NM_000527.4:c.1846-2A>C	*LDLR*
6.	16	F	280	200	50	102	NM_000527.4:c.2416dupG	*LDLR*
							NM_000384.2:c.10708C>T	*APOB*
7A.	5	F	278	170	61	110	NM_000527.4:c.373_379delCAGTTCG	*LDLR*
7B.	10	M	426	348	56	82	NM_000527.4:c.373_379delCAGTTCG	*LDLR*
8.	12	M	318	237	67	73	NM_000527.4:c.190+1G>A	*LDLR*
9A.	13	M	303	237	49	137	NM_000527.4:c.1775G>A	*LDLR*
9B.	17	F	289	218	39	162	NM_000527.4:c.1775G>A	*LDLR*
10.	8	F	265	165	73	138	NM_022436.2:c.1285G>A	*ABCG5*
11.	13	M	218	156	48	121	NM_000527.4:c.1775G>A	*LDLR*
12A.	13	F	234	171	40	90	*-*	
12B.	14	F	233	191	42	102	*-*	
13.	10	M	193	135	45	NA	-	
14.	10	M	279	150	44	138	NM_000237.2:c.106G>A	*LPL*
15.	14	F	336	267	52	106	-	
16.	10	M	239	174	53	125	-	
17.	9	F	252	165	53	47	NM_000527.4:c.1775G>A	*LDLR*
18.	4	F	266	200	44	111	NM_000384.2:c.10580G>A	*APOB*
19.	9	F	283	201	69	65	-	
20.	9	F	290	188	64	188	-	
21.	6	F	219	137	76	50	-	
22.	2	F	278	216	25	546	-	
23.	5	M	307	240	52	73	NM_000527.4:c.1775G>A	*LDLR*
24.	13	F	NA	NA	NA	NA	NM_000527.4:c.284G>T	*LDLR*
25.	6	F	348	263	NA	NA	NM_000527.4:c.1775G>A	*LDLR*
26.	16	F	440	367	43	144	-	
27.	17	F	380	303	61	81	NM_000527.4:c.1775G>A	*LDLR*
28.	4	M	265	198	47	100	-	
29.	9	F	233	143	56	178	-	
30.	9	F	402	316	68	84	NM_000527.4:c.530C>T	*LDLR*
31.	5	F	345	273	55	86	NM_000527.4:c.190+1G>A	*LDLR*
32.	14	M	320	263	54	92	NM_000527.4:c.1775G>A	*LDLR*
							NM_000237.2:c.953A>G	*LPL*
33.	13	M	331	245	74	62	NM_000527.4:c.1916T>A	*LDLR*
34.	10	F	350	310	42	78	NM_000527.4:c.324_325delGTinsTC	*LDLR*
35.	11	M	459	398	42	83	NM_000527.4:c.2063delA	*LDLR*
36.	3	M	314	252	44	80	NM_000527.4:c.1061A>G	*LDLR*
37.	15	F	364	294	45	124	NM_000527.4:c.324_325delGTinsTC	*LDLR*
38.	5	F	745	693	37	73	NM_000527.4:c.1747C>T	*LDLR*
39.	10	F	331	247	73	54	-	
40.	9	F	288	216	57	74	NM_000527.4:c.798T>A	*LDLR*
41.	13	F	387	316	53	86	NM_000527.4:c.1433G>A	*LDLR*
42.	14	F	274	196	66	63	NM_000384.2:c.10580G>A	*APOB*
43.	12	F	290	206	59	80	-	
44.	17	M	280	262	49	97	NM_000527.4:c.1061A>G	*LDLR*
45.	9	M	265	144	73	95	NM_000527.4:c.1775G>A	*LDLR*
46.	2	F	228	170	48	52	NM_000384.2:c.10580G>A	*APOB*
47.	6	F	248	177	68	103	-	
48.	8	F	230	185	42	106	-	
49.	3	M	209	101	77	74	NM_022436.2:c.1336C>T	*ABCG5*
50.	15	F	229	170	46	109	-	
51.	10	M	219	177	16	72	-	
52.	10	M	242	170	46	135	-	
53.	6	F	235	154	NA	49	NM_000527.4:c.907C>T	*LDLR*

NA—not applicable.

**Table 2 genes-13-00999-t002:** Mean concentration of each lipid fraction in the following groups of patients.

	TC [mg/dL] Mean ± SD	LDL [mg/dL] Mean ± SD	HDL [mg/dL] Mean ± SD	TG [mg/dL] Mean ± SD
Study group	296.4 ± 85.2	224.0 ± 89.2	52.4 ± 12.7	104.4 ± 70.4
Patients with genetic findings	329.1 ± 92.8	257.4 ± 98.1	51.6 ± 9.8	90.6 ± 30.4
Patients without genetic findings	245.1 ± 31.6	175.8 ± 31.4	53.5 ± 15.9	126.2 ± 102.5

**Table 3 genes-13-00999-t003:** The variant table for selected aberrations identified by NGS sequencing, including the name of the affected gene with exon (indicated in underline)/intron localization, HGVSc and HGVSp description, genotype, variant type, in silico predictions, initial classification and number of the identified patients.

Gene	Exon /Intron	Hgvsc	Hgvsp	Genotype	Variant Type	In Silico Prediction			Classification	Nr. of Cases
						Polyphen	SIFT	Mutation Taster		
*LDLR*	12	NM_000527.4:c.1775G>A	NP_000518.1:p. Gly592Glu	het	missense	probably damaging (0.927)	deleterious (0.01)	disease causing	pathogenic	9
*LDLR*	4	NM_000527.4:c.373_ 379delCAGTTCG	NP_000518.1:p. Gln125SerfsTer79	het	deletion	NA	NA	disease causing	presumed pathogenic	2
*LDLR*	2	NM_000527.4:c.190+1G>A	NA	het	splice donor	NA	NA	NA	presumed pathogenic	2
*LDLR*	4	NM_000527.4:c.324_ 325delGTinsTC	NP_000518.1:p. Cys109Arg	het	missense	NA	NA	NA	presumed pathogenic	2
*LDLR*	8	NM_000527.4:c.1061A>G	NP_000518.1:p. Asp354Gly	het	missense	probably damaging (1)	deleterious (0)	disease causing	presumed pathogenic	2
*LDLR*	17	NM_000527.4:c.2416dupG	NP_000518.1:p. Val806GlyfsTer11	het	duplication	NA	NA	NA	presumed pathogenic	1
*LDLR*	12	NM_000527.4:c.1846-2A>C	NA	het	splice acceptor	NA	NA	NA	presumed pathogenic	1
*LDLR*	3	NM_000527.4:c.284G>T	NP_000518.1:p. Cys95Phe	het	missense	probably damaging (0.999)	deleterious (0)	disease causing	presumed pathogenic	1
*LDLR*	4	NM_000527.4:c.530C>T	NP_000518.1:p. Ser177Leu	het	missense	probably damaging (1)	deleterious (0.01)	disease causing	presumed pathogenic	1
*LDLR*	13	NM_000527.4:c.1916T>A	NP_000518.1:p. Val639Asp	het	missense	benign (0.245)	deleterious (0)	disease causing	presumed pathogenic	1
*LDLR*	14	NM_000527.4:c.2063delA	NP_000518.1:p. Asn688ThrfsTer21	het	deletion	NA	NA	NA	presumed pathogenic	1
*LDLR*	12	NM_000527.4:c.1747C>T	NP_000518.1:p. His583Tyr	het	missense	probably damaging (1)	deleterious (0)	disease causing	presumed pathogenic	1
*LDLR*	5	NM_000527.4:c.798T>A	NP_000518.1:p. Asp266Glu	het	missense	probably damaging (0.979)	deleterious (0)	disease causing	presumed pathogenic	1
*LDLR*	10	NM_000527.4:c.1433G>A	NP_000518.1:p. Gly478Glu	het	missense	probably damaging (0.998)	deleterious (0)	disease causing	presumed pathogenic	1
*LDLR*	6	NM_000527.4:c.907C>T	NP_000581.1:p. Arg303Trp	het	missense	probably damaging (0.998)	Tolerated (0.37)	disease causing	presumed pathogenic	1
Sum of *LDLR*-positive patients										27
*APOB*	26	NM_000384.2:c.10580G>A	NP_000375.2:p. Arg3527Gln	het	missense	probably damaging (0.93)	NA	disease causing	pathogenic	3
*APOB*	26	NM_000384.2:c.10708C>T	NP_000375.2:p. His3570Tyr	het	missense	benign (0.003)	NA	polymorphism	uncertain significance	1
Sum of *APOB*-positive patients										4
*ABCG5*	5	NM_022436.2:c.593G>A	NP_071881.1:p. Arg198Gln	het	missense	probably damaging (1)	deleterious (0)	disease causing	uncertain significance	1
*ABCG5*	9	NM_022436.2:c.1285G>A	NP_071881.1:p. Ala429Thr	het	missense	possibly damaging (0.558)	deleterious (0)	polymorphism	uncertain significance	1
Sum of *ABCG5*-positive patients										2
*LPL*	2	NM_000237.2:c.106G>A	NP_000228.1:p. Asp36Asn	het	missense	benign (0.066)	tolerated (0.16)	disease causing	risk factor	1
*LPL*	6	NM_000237.2:c.953A>G	NP_000228.1:p. Asn318Ser	het	missense	benign (0.019)	tolerated (0.24)	disease causing	presumed pathogenic	1
Sum of *LPL*-positive patients										2
Total sum of NGS-positive patients										35

NA—not applicable.

**Table 4 genes-13-00999-t004:** Lipid concentration values of described siblings and their parents.

Subject	Age	TC	LDL	HDL	TG	c.373_379delCAGTTCG Presence
		mg/dL				
Child 1	10	426	348	56	82	+
Child 2	5	278	170	61	110	+
Mother	41	266	132	97	186	-
Father	43	384	317	51	83	+

## Data Availability

The data of novel variant described in the manuscript were submitted to the ClinVar database with the accession number VCV001300030.1.

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
