# Peer review of "Identification of New Genetic Determinants in Pediatric Patients with Familial Hypercholesterolemia Using a Custom NGS Panel"

_genes, 2022, doi:10.3390/genes13060999_

Round 1

Reviewer 1 Report

This manuscript is an NGS study of FH genes in a paediatric cohort, with the discovery of a new mutation.

-The introduction is too long, please cut-down; the authors mention that the STAP1 and LDLRAP1 gene could be associated with FH, there are also rare APOE that are associated with FH

-In a paediatric cohort, you would expect the phenotype to be more discriminatory than in adults, please work this into your study rationale

-Please end your introduction with the aims and hypothesis to the study

-Explain your rational for choosing the SB criteria, ie. the DLCN criteria does not apply for children

-Methods: ethics approval? and location of the study/recruitment? also specify your inclusion criteria in methods (not results)

-Some of the large tables could be supplementary material instead, if journal allows

-What about polygenic risk scores and Lp(a) levels and SNPs in this cohort?

-The discussion is also too long and not well structured; any limitations to discuss?

-Can the authors discuss the implementation and cost-effectiveness of NGS in Poland?

Attention to English writing and gramma recommended throughout the entire manuscript.

Reviewer 2 Report

The authors analyzed variants in genes related to lipid metabolism in pediatric patients with familial hypercholesterolemia. The study is interesting as usually only the most common genetic mutations are checked in patients. The strength of the study is the search for potentially pathogenic FH-related variants. The manuscript is well written and clear.

I have some minor concerns:

1.       Did you use for patients without functionally significant point substitutions MLPA to find possible bigger mutations in the LDLR promoter and exons (deletions and/or duplications)?

2.       The discussion is short and a lot more can be fleshed out, particularly about new FH-related genes.
